# Uncertainties in the Geostationary Ocean Color Imager (GOCI) Remote Sensing Reflectance for Assessing Diurnal Variability of Biogeochemical Processes

**Javier Concha** [1,2,*], **Antonio Mannino** [1], **Bryan Franz** [1] **and Wonkook Kim** [3]

[1]   Ocean Ecology Lab, NASA Goddard Space Flight Center, Greenbelt, MD 20771, USA;
     antonio.mannino@nasa.gov (A.M.); bryan.a.franz@nasa.gov (B.F.)
[2]   Universities Space Research Association, Columbia, MD 21046, USA
[3]   Korea Institute of Ocean Science and Technology, Busan 49111, Korea; wkkim@kiost.ac.kr
*   Correspondence: jaconcha@gmail.com

**Abstract:** Short-term (sub-diurnal) biological and biogeochemical processes cannot be fully captured by the current suite of polar-orbiting satellite ocean color sensors, as their temporal resolution is limited to potentially one clear image per day. Geostationary sensors, such as the Geostationary Ocean Color Imager (GOCI) from the Republic of Korea, allow the study of these short-term processes because their orbit permit the collection of multiple images throughout each day for any area within the sensor's field of regard. Assessing the capability to detect sub-diurnal changes in in-water properties caused by physical and biogeochemical processes characteristic of open ocean and coastal ocean ecosystems, however, requires an understanding of the uncertainties introduced by the instrument and/or geophysical retrieval algorithms. This work presents a study of the uncertainties during the daytime period for an ocean region with characteristically low-productivity with the assumption that only small and undetectable changes occur in the in-water properties due to biogeochemical processes during the daytime period. The complete GOCI mission data were processed using NASA's SeaDAS/l2gen package. The assumption of homogeneity of the study region was tested using three-day sequences and diurnal statistics. This assumption was found to hold based on the minimal diurnal and day-to-day variability in GOCI data products. Relative differences with respect to the midday value were calculated for each hourly observation of the day in order to investigate what time of the day the variability is greater. Also, the influence of the solar zenith angle in the retrieval of remote sensing reflectances and derived products was examined. Finally, we determined that the uncertainties in water-leaving "remote-sensing" reflectance ($R_{rs}$) for the 412, 443, 490, 555, 660 and 680 nm bands on GOCI are $8.05 \times 10^{-4}$, $5.49 \times 10^{-4}$, $4.48 \times 10^{-4}$, $2.51 \times 10^{-4}$, $8.83 \times 10^{-5}$, and $1.36 \times 10^{-4}$ sr$^{-1}$, respectively, and $1.09 \times 10^{-2}$ mg m$^{-3}$ for the chlorophyll-*a* concentration (Chl-*a*), $2.09 \times 10^{-3}$ m$^{-1}$ for the absorption coefficient of chromophoric dissolved organic matter at 412 nm ($a_g$ (412)), and 3.7 mg m$^{-3}$ for particulate organic carbon (POC). These $R_{rs}$ values can be considered the threshold values for detectable changes of the in-water properties due to biological, physical or biogeochemical processes from GOCI.

**Keywords:** Geostationary Ocean Color Imager (GOCI); ocean color; diurnal dynamics; diurnal variability; uncertainties

## 1. Introduction

Ocean waters are highly dynamic due to environmental factors such as heating of the surface ocean layer, fluctuation in wind intensity, surface currents, tidal cycles, changes in vertical mixing layers and variation of sunlight radiation. These dynamics produce changes in marine ecosystem processes, such as ocean primary production, carbon stocks, export production and phytoplankton community composition and their effects can be measured at different time scales, from decades to years, all the way to days or hours. Longer term variations, such as seasonal, interannual and decadal patterns in phytoplankton stocks, optical properties and primary production, have been extensively studied using low earth orbit (LEO) assets (e.g., [1–4]) such as the Sea-Viewing Wide Field-of-View Sensor (SeaWiFS) [5], Moderate Resolution Imaging Spectroradiometer (MODIS) [6] and the Visible Infrared Imaging Radiometer Suite (VIIRS) [7]). However, these ocean color-enabled sensors do not have the temporal resolution needed to capture short-term (sub-diurnal) dynamics.

Ocean color sensors in geostationary orbit (GEO) can provide a means to better understand ocean processes that vary at sub-diurnal (or day-to-day) scale because of their multiple times per day acquisition capability [8,9]. The Republic of Korea's Geostationary Ocean Color Imager (GOCI), launched in 2010, is the first and only (to date) operational geostationary ocean color sensor [10] and it has proven to be capable of detecting sub-diurnal variations of the coastal waters in Korea. A variety of algorithms have been developed that utilize GOCI spectral observations to retrieve water-column constituents, including concentrations of the phytoplankton pigment chlorophyll-*a* (Chl-*a*) and total suspended material (TSM) [11–13]. GOCI data have also been used to determine the diurnal submesoscale variability of turbidity fronts [14], internal waves [15], red tides and green algae [16,17]. The success of GOCI has prompted the development of future GEO missions such as GOCI-II, scheduled to be launched in 2019, and formulation studies on a European geostationary satellite Ocean Color Advanced Permanent Imager (GEO-OCAPI). NASA has conducted and recently concluded pre-formulation studies for the Geostationary Coastal and Air Pollution Events (GEO-CAPE) mission. Taken together, GEO missions such as these three have the capability to provide quasi-global coverage at low and mid-latitudes [9].

Sub-diurnal processes in oligotrophic waters are of vital importance for the balance in the Earth system. Previous studies have reported diel variations for several oceanic microorganisms (e.g., phytoplankton and bacteria) present in these types of waters from in situ [18–21] and lab measurements [22–24]. The variations in the properties of these organisms, such as abundance or composition, affect the bulk optical properties of seawater [18,25–28], and therefore, such diel variations could be potentially detected by optical instruments. However, changes in biogeochemical stocks and rates and thus diel variability in optical properties in such oligotrophic ocean regions tend to be very small and can potentially fall within the uncertainty levels of current satellite instrumentation and processing algorithms. For instance, Claustre et al. (2008) [29] reported a total daily increase in vertically integrated POC of 0.2 g m$^{-2}$ over the top 80 m or equivalent to 2.5 mg m$^{-3}$, for the South Pacific Gyre during mid-November 2004. This level of change is too low to be detected by ocean color sensors just from the uncertainty in the POC algorithms [30].

Therefore, in order to determine whether sub-diurnal and day-to-day differences in GOCI-derived optical and biogeochemical ocean properties are related to real physical, ecological and biogeochemical processes, the levels of uncertainties of GOCI data products must first be assessed. Hence, the primary objective of this study is to quantify the inherent uncertainties of GOCI remote sensing reflectances ($R_{rs}$) and derived products when assessing diurnal variability. First, we processed GOCI data to $R_{rs}$ over a clear water region, which is assumed to express little to no diurnal and day-to-day variability due to biology or physical processes. To determine the validity of this assumption, the absence of variability from sub-diurnal to multiple day timescales is investigated. We verified that this assumption holds true at these timescales for our study region. Next, we estimated GOCI $R_{rs}$ and derived biogeochemical product uncertainties within the region of study using two different approaches: (1) an analysis of daily statistics, specifically the daily standard deviation and the percentage coefficient of variation (CV), to

estimate the deviation from the daily mean of the overall mission and (2) the relative difference with respect to a midday value ($R\Delta_t[\%]$) to estimate the deviation from a midday value by time of the day. Also, the effect of the solar zenith angle (SZA), which is convolved with the bidirectional reflectance distribution function (BRDF) effect [31], on the ocean color products retrieval was investigated. Our findings suggest that diurnal variability is discernible with GOCI within a certain level of uncertainties and that there does not appear to be a considerably negative impact from the sensor-solar geometry in the algorithms.

## 2. Data and Sensor Characteristics

This study focused on ocean color data from GOCI over a specific open ocean region within its coverage area. GOCI's specifications and a description of the study area are described in this section.

### 2.1. GOCI Data

GOCI, which was launched on 26 June 2010, monitors the Northeast Asian waters surrounding the Korean peninsula, generating eight images per day (from 00:15 Greenwich Mean Time (GMT) to 07:45 GMT at one hour interval or from 9:15 to 16:45 hours local time) with a spatial resolution of 500 m at 130°E and 36°N. It covers an area of about 2500 km × 2500 km. It has eight spectral bands (6 bands in the visible: 412, 443, 490, 555, 660 and 680 nm; 2 bands in the near infrared (NIR): 745 and 865 nm). GOCI operates in a 2D staring-frame capture mode in a geostationary orbit onboard the Communication Ocean and Meteorological Satellite (COMS) of the Republic of Korea. The data acquisition over the observational coverage area of GOCI is accomplished with a step-and-stare method that takes 16 step-by-step slots by the scan of a pointing mirror with a dedicated CMOS detector array (1432 × 1415 pixels) [32].

The images used in this analysis span from the beginning of GOCI's mission (May 2011) until January 2018, resulting in a total of about 20,000 images. The GOCI Level-1B calibrated top-of-atmosphere (TOA) radiance data were obtained from the Ocean Biology Distributed Active Archive Center (OB.DAAC) at the NASA's Goddard Space Flight Center, maintained by the Ocean Biology Processing Group (OBPG). The OB.DAAC acts as a mirror site for the GOCI data provided by the Korea Ocean Satellite Center of the Korea Institute of Ocean Science and Technology. These data are freely available for direct download from the OB.DAAC (https://oceancolor.gsfc.nasa.gov/).

### 2.2. Area of Study

The area of study is located south of Japan (Figure 1), in the northwestern fringe of the North Pacific Subtropical Gyre (NPSG) in a transition region between its western boundary current, that is, the Kuroshio Current and the subtropical countercurrent (STCC) [33]. As described in the Introduction section, biological and biogeochemical processes in this study area produce different levels of variability at different temporal scales. For instance, the NPSG gyre expands or contracts depending on the season following the seasonal strength of the winds and convective upper-ocean mixing [34,35]. This behavior in the NPSG may lead to this region changing seasonally from oligotrophic to mesotrophic conditions. For instance, the Chl-*a* concentration in the study region ranges from 0.05 mg m$^{-3}$ to 0.2 mg m$^{-3}$ (as described in Section 4.1), depending on the season, a range comparable to the one reported by [35] with a mean satellite-derived Chl-*a* concentration in the NPSG gyre ranging between 0.07 mg m$^{-3}$ and 0.11 mg m$^{-3}$. Furthermore, because the study region is at the limit of NPSG, different types of physical forcing than the ones affecting the gyres may affect it. For instance, fronts or winds may induce upwelling, providing nutrients to the euphotic zone. Combined with the fact that the nutricline is much shallower along the fringes of the gyre than in its interior, this can contribute to greater phytoplankton growth compared to within the gyres.

This area of study is referred to as the GCWS (GOCI Clear Water Subset) region hereafter, with boundaries: north = 28.4950°, south = 26.0960°, west = 137.3380° and east = 142.0920°, centered at 27.33°N and 139.71°E. This area is the same area used by [36] for obtaining the updated vicarious

calibration gains for GOCI. The GCWS region is approximately 433 × 968 GOCI pixels, equivalent to approximately 100,000 km². Despite the dynamics previously discussed, the GCWS was selected because of the assumption that the physical (e.g., solar heating, wind, waves) and biogeochemical processes (e.g., $CO_2$ fixation) occurring in this region will have a small and thus, undetectable effect on short term (sub-diurnal) variability of the ocean constituents and thus optical properties. In this manner, the variability in the GOCI-derived products due to changes in the in-water constituents will be minimized and therefore, the variability introduced by sensor radiometric uncertainty (e.g., noise, systematic error), viewing geometry and algorithm can be quantified. For this region, the range of SZA during the acquisition time varies between 0° and 90° through the year and from approximately 29° to 37° for the sensor (viewing) zenith angle. The boundaries of the GCWS region were selected to be inside the slot located in the lower right corner of the L1B image (slot number 13 in Reference [37]) to avoid the Stray-Light-Driven Interslot Radiometric Discrepancy (ISRD) at the near-boundary of the interslot areas [37,38]. It has been estimated that the stray-light-driven radiometric anomalies could reach up to 20% in some bands and therefore, the potentially affected areas were excluded in this study.

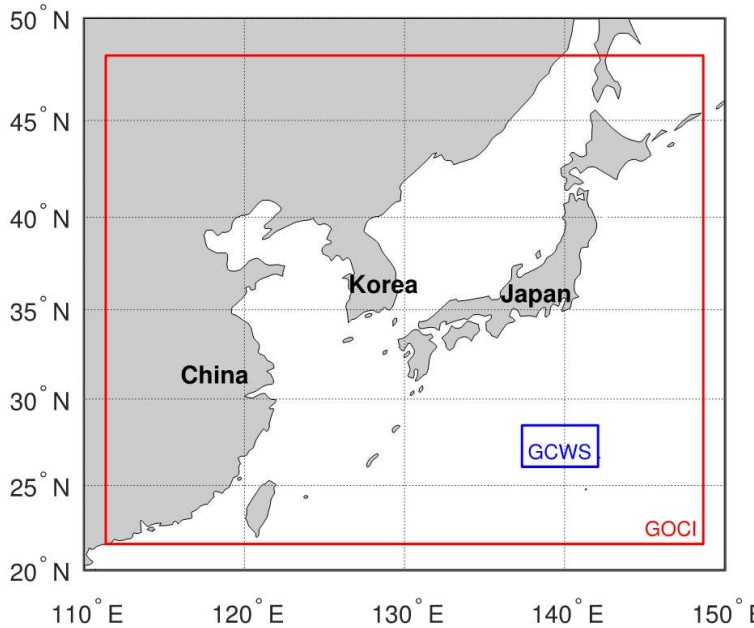

**Figure 1.** The study area (GCWS region; blue box) is located over oligotrophic waters to the south of Japan within the coverage area of GOCI (red box). The GCWS region covers 433 × 968 pixels, which is equivalent to 100,000 km².

## 3. Processing Approach

The analysis of the uncertainties is made over the GOCI-derived Level-2 (L2) products that include remote-sensing reflectance, chlorophyll-*a* concentration (Chl-*a*), chromophoric dissolved organic matter (CDOM) absorption coefficient at 412 nm ($a_g(412)$) and particulate organic carbon (POC). After processing the data to L2, these data were screened for quality assurance.

### 3.1. Conversion to Level 2

GOCI geolocated and radiometrically calibrated (Level-1B) data (L1B) were processed to Level-2 biogeophysical products (L2) using the multi-sensor Level-1 to Level-2 generator (l2gen) version 9.2.0-V2017.0.3 distributed with the SeaWiFS Data Analysis System (SeaDAS) (http://seadas.gsfc.nasa.gov/). The l2gen code reads Level-1B observed top-of-atmosphere (TOA) radiances, applies one of the atmospheric correction schemes available and outputs various products such as radiances or reflectances (e.g., spectral remote-sensing reflectance, $R_{rs}(\lambda)$) and derived biogeophysical parameter (e.g., chlorophyll-*a* concentration). As part of the l2gen processing, each pixel is masked with different

flags that reflect warnings or errors generated during the processing to assure the quality of the data [39].

The atmospheric correction scheme applied to this study was the default algorithm [31] for GOCI (aer_opt = −2) that uses an estimation of the aerosol contribution described by Gordon & Wang (1994) [40], including a near infrared (NIR) iterative correction by Bailey et al. (2010) [41], a suite of aerosol models developed by Ahmad et al. (2010) [42] with selection that is dependent on relative humidity (RH) and the spectral slope observed in two NIR channels and a BRDF correction described by Morel et al. (2002) [43]. GOCI's two near infrared (NIR) bands at 745 and 865 nm were used for the aerosol model selection. This atmospheric correction approach assumes a plane-parallel geometry, ignoring earth curvature, for the vector radiative transfer simulations used for the computation of the look-up tables of Rayleigh and aerosol reflectance. A vicarious calibration specific for GOCI was applied, based on match-ups with MODIS-Aqua over the same GCWS region [36].

### 3.2. Data Screening

For the analysis of uncertainties described in the following sections, we chose to use a single value that represents each L2 product: the filtered mean. To ensure a good quality of the data used for the analysis, an exclusion criterion (filtering) that is based on [39] was applied for the calculation of this filtered mean (Figure 2). One of the goals of this filtering is to ensure the removal of extreme short term variability from events that are not the focus of this work such as episodic storms or dust clouds. In order to avoid the effect of outliers in the calculations, the following screening criteria were applied for selecting the pixels within the GCWS region to be used for the calculation of the filtered mean:

$$(\text{Med} - 1.5\sigma) < X_i < (\text{Med} + 1.5\sigma) \tag{1}$$

where $X_i$ is the $i$th filtered pixel within the GCWS region, Med is the median value of the unflagged pixels and $\sigma$ is the standard deviation of the unflagged pixels. Then, the filtered mean was calculated as:

$$\text{Filtered Mean} = \frac{\sum_i^{\text{NFP}} X_i}{\text{NFP}} \tag{2}$$

where NFP is the Number Filtered Pixels, that is, the number of unflagged values within Med $\pm$ 1.5$\sigma$. Note the difference with Equation (4) in Reference [39], in which the mean of the unfiltered data was used instead of the median, as in this case. The use of the median value for the calculation of the filtered mean minimizes the influence of outliers.

A coefficient of variation ($CV$), which is defined as the standard deviation divided by the mean, is calculated for the $R_{\text{rs}}$ in the blue and green bands and the aerosol optical thickness at 865 nm products (i.e., a mean and standard deviation is calculated for the $433 \times 968 = 419{,}144$ pixels of each $R_{\text{rs}}(412)$, $R_{\text{rs}}(440)$, $R_{\text{rs}}(490)$, $R_{\text{rs}}(555)$ and AOT(865) products) [39]. Then, the median of all these CV values for each L2 file is recorded (Med[$CV$]). Next, only L2 products with an associated Med[$CV$] smaller than 0.25 (25%) were used for the uncertainties analysis. Also, to provide statistical confidence in the filtered mean values, NFP is required to be at least a third of the number of total pixels in the GCWS region (i.e., NFP $\geq$ NTP/3 = 139,714) for the L2 product to be considered in the analysis. This is equivalent to stating that at least a third of the area of the GCWS region has valid pixel values associated with it. Both of these thresholds, the minimal valid area and the maximum Med[$CV$], were determined through a trial-and-error method that maximized a tradeoff between sufficiency of data values for statistical robustness and the influence of outliers.

Additionally, we excluded L2 products with solar and sensor zenith of the center pixel that exceeded 75° and 60°, respectively, to avoid extreme solar and viewing geometries [39]. 1600 files of a total of 20,834 (7.6%) exceeded the SZA threshold. The sensor zenith angle is between 29° and 37° for this study region and therefore, this criterion did not exclude any pixels.

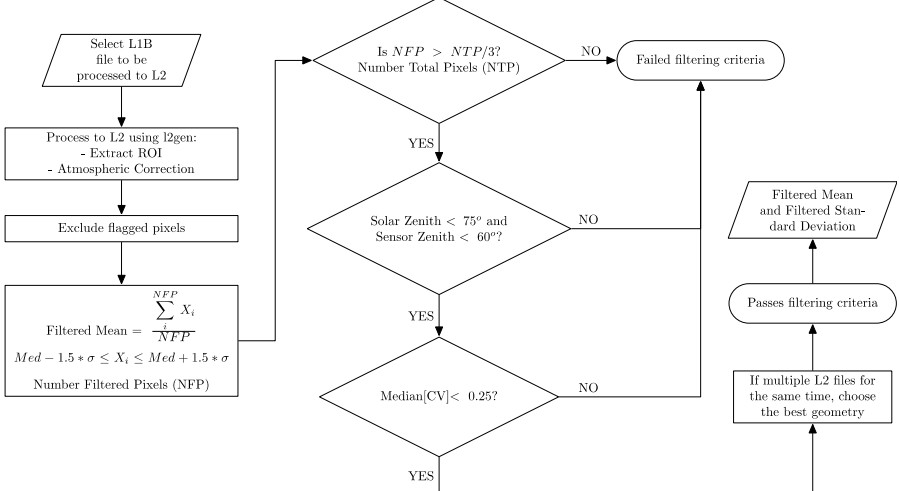

**Figure 2.** Flow diagram of the methodology employed to calculate the filtered mean and standard deviation for each L2 product including the exclusion criteria applied to ensure good data quality.

*3.3. Bio-Optical Algorithms*

In addition to the $R_{rs}$ products, three biogeochemical products were used to study uncertainties in diurnal variability using GOCI. Two of these were default global algorithms found in SeaDAS/l2gen and the third one is a CDOM absorption retrieval algorithm currently under evaluation.

3.3.1. Chlorophyll-*a* Concentration (Chl-*a*)

The standard Chl-*a* product produced by the OBPG blends two algorithms. The maximum band ratio algorithm (OCx) relies on empirically derived relationships that statistically relate in situ pigment concentration with field-measured band ratios of remote sensing reflectance, $R_{rs}(\lambda)$, of blue and green bands [44]. This algorithm is updated regularly to include the most recent field measurements. OBPG recently adopted the color index (CI) Chl-*a* algorithm of Hu et al. (2012) [45], a three-band difference algorithm, to compute Chl-*a* within clear waters. OBPG generates a single Chl-*a* product (as the standard Chl-*a* product) using both OCx and CI algorithms, where CI-derived values are applied when Chl-*a* < 0.15 mg m$^{-3}$ and OCx when Chl-*a* is > 0.2 mg m$^{-3}$. Weighted Chl-*a* values are computed for the interval between these values to assure a smooth transition for the merged data product. The blended algorithm is commonly referred to as OCI [45].

Briefly, the *CI* algorithm for GOCI has the following form:

$$CI = R_{rs}(555) - \left[ R_{rs}(443) + \frac{(555-443)}{(660-443)} \times [R_{rs}(660) - R_{rs}(443)] \right]$$
$$\text{Chl-}a = 10^{0.4909+191.6590 \times CI}, \; CI \leq -0.0005 \text{ sr}^{-1} \tag{3}$$

and the standard OCx algorithm has the form:

$$log_{10}(\text{Chl-}a) = a_0 + \sum_{i=1}^{4} a_i \left[ log_{10} \left( \frac{R_{rs}(\lambda_{\text{blue}})}{R_{rs}(\lambda_{\text{green}})} \right) \right]^i \tag{4}$$

where the coefficients $a_0, \ldots, a_4$ are sensor specific. For GOCI, the 3-band version of OCx (OC3) is used, with the 443, 490, 555 nm bands and the coefficients $a_0 = 0.2515$, $a_1 = -2.3798$, $a_2 = 1.5823$, $a_3 = -0.6372$ and $a_4 = -0.5692$.

### 3.3.2. Particulate Organic Carbon (POC)

The standard algorithm to retrieve the concentration of particulate organic carbon (POC) is based on an empirical relationship between in situ POC measurements and blue-to-green band ratios of $R_{rs}$ [46]. This algorithm uses the 443 and 555 nm bands for GOCI:

$$\mathrm{POC} = 203.2 \times \left[ \frac{R_{rs}(443)}{R_{rs}(555)} \right]^{-1.034} \tag{5}$$

### 3.3.3. Chromophoric Dissolved Organic Matter Absorption Coefficient at 412 nm ($a_g(412)$)

Mannino et al. (2014) [47] developed an algorithm for the retrieval of chromophoric dissolved organic matter (CDOM) absorption at 412 nm ($a_g(412)$) spanning eutrophic to oligotrophic waters along the northeastern U.S. coast. This algorithm was initially implemented for SeaWiFS and MODIS Aqua and now it is included in l2gen as ag_412_mlrc for testing. It is based on field measurements collected throughout the continental margin of the northeastern U.S. from 2004 to 2011. This algorithm involves a least squares linear regression of $a_g(\lambda)$ with multiple $R_{rs}$ bands within a multiple linear regression (MLR) analysis. The bands used in this case are the 443 and 555 nm bands. This algorithm takes the following form:

$$
\begin{aligned}
Y &= -2.784 - 1.146 \times Ln(R_{rs}(443)) + 1.008 \times Ln(R_{rs}(555)) \\
a_g(412) &= e^Y
\end{aligned} \tag{6}
$$

## 4. Results and Discussion

### 4.1. Seasonality

An analysis of the complete GOCI time series in the GCWS region was conducted to provide an understanding of the long term variability (seasonal to interannual) in $R_{rs}$ and biogeochemical products to provide a context for interpreting the short term variability. The time series of $R_{rs}$ exhibit an expected seasonality with a recurrent pattern for all years, more evident in the blue bands (Figure 3), due to biological and biogeochemical processes occurring in the area of study. This is corroborated with the chl-*a*, $a_g(412)$ and POC products (Figure 4). The chl-*a* seasonal signal suggests an increase in phytoplankton biomass and productivity from winter through early spring. The peak of the chl-*a* time series in early spring resembles the behavior of the phytoplankton growth occurring in the North Pacific Subtropical Gyre (NPSG) [35]. The time-scale of these biological and biogeochemical processes occurring in the GCWS region is much longer and gradual than the time-scale of interest for this study.

These ocean color time series were created with filtered mean values and for each day there are potentially eight values of observations, which represents the diurnal variability and explains the daily distribution of the data in Figures 3 and 4. The observations from the early part of the mission (before 05-15-2011) were collected during the in-orbit test period of the mission and were not included in the following set of analyses. The histograms for the blue bands at 412 and 443 nm (Figure 3a,b) exhibit a bimodality due to the seasonal variability of the phytoplankton and possibly CDOM. This behavior is reflected in the biogeochemical products (Figure 4a–c). Also, there are more valid data points in the summer-fall period than the winter-spring one as one would expect for this region due to cloud cover, extreme SZA or atmospheric correction failure.

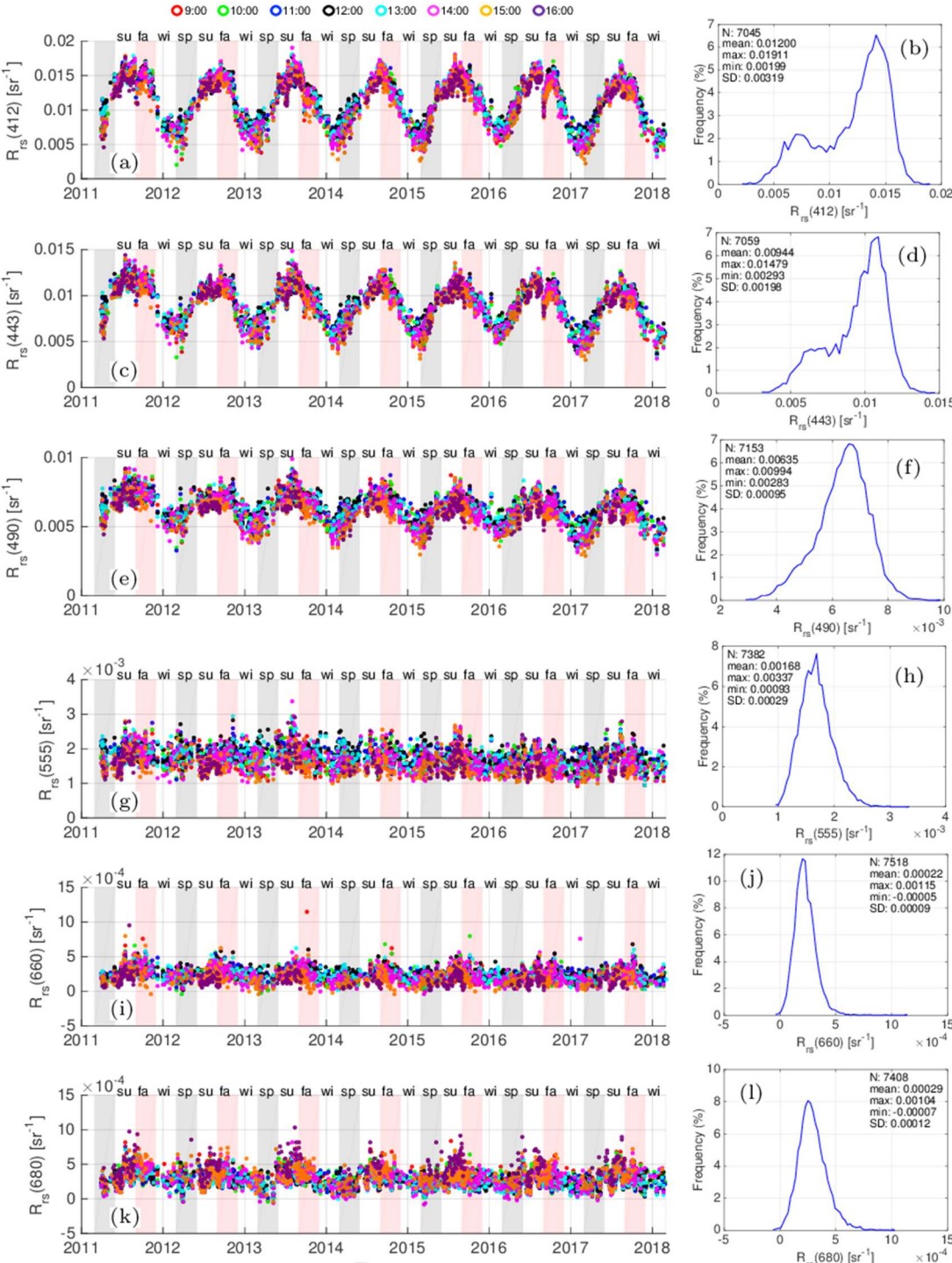

**Figure 3.** Time Series and histograms of remote sensing reflectance ($R_{rs}(\lambda)$) (**a**,**b**) $R_{rs}(412)$, (**c**,**d**) $R_{rs}(443)$, (**e**,**f**) $R_{rs}(490)$, (**g**,**h**) $R_{rs}(555)$, (**i**,**j**) $R_{rs}(660)$, (**k**,**l**) $R_{rs}(680)$ products for the GCWS region. The complete GOCI mission was processed to Level-2 and a filtered mean was calculated for each image over the GCWS region. The data are color coded by time of day in local time. Labels are summer (su), fall (fa), winter (wi) and spring (sp). The histograms show the total number (N), mean, maximum, minimum and standard deviation (SD) of the values that passed the exclusion criteria.

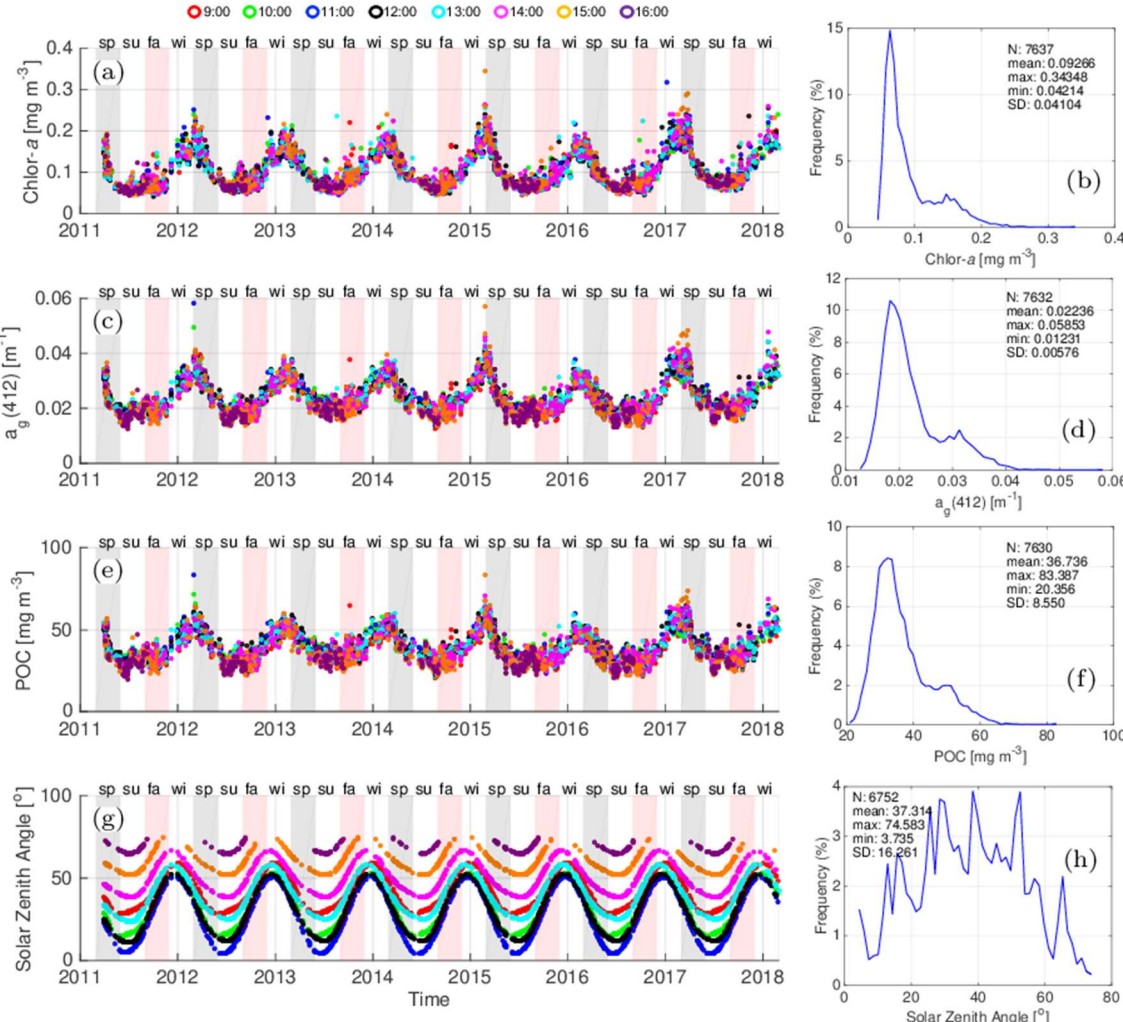

**Figure 4.** Time Series and histograms for the (**a**,**b**) Chl-*a*, (**c**,**d**) $a_g$(412), (**e**,**f**) POC products and (**g**,**h**) solar zenith angle for the GCWS region. The data are color coded by time of day. See Figure 3 for additional details.

## 4.2. Diurnal and Day-to-Day Variability

The primary assumption for this work is that the in-water constituents within the GCWS region remain temporally and spatially homogeneous over short periods of time, that is, the optical properties of the water do not change considerably during the daytime nor from day-to-day due to physical and/or biogeochemical processes. Specifically, any sub-diurnal to day-to-day variability in the GCWS will be too small to be measurable by GOCI. Here, statistical analyses on diurnal and three-day sequences were performed in order to test this assumption. Also, the diurnal statistics computed here provide an estimation of the threshold uncertainties for GOCI data products, which can be applied in more dynamic regions to quantify short term changes in $R_{rs}$ and derived biogeochemical data products.

First, one three-day sequence is provided as an example and then, results are shown from all valid three-day sequences used to obtain a quantitative estimate of the sub-diurnal to day-to-day variability. Under ideal circumstances, if the water is temporally and spatially homogeneous, we would expect that all the values during the day to be the same and the values for all the three-day sequences to be the same too, at least within the uncertainty of the satellite sensor calibration and algorithms applied. This assumes that the atmospheric correction algorithm is properly compensating for changes in aerosol and atmospheric gas composition, solar and viewing geometry that influence atmospheric radiant

path reflectance, surface reflection/refraction effects and the bidirectional reflectance of the subsurface light field [31].

Out of the 2500 days within the entire GOCI mission investigated, there are only 96 three-day sequences with valid values for the all the bands and for all times of the days. Given the cloudy nature of the region and the Earth in general, the identification of 96 complete three-day diurnal sequences and many diurnal sequences missing only a few hourly observations supports the applicability of such observations from geostationary orbit to study ocean processes in more dynamic areas. As an example, a three-day sequence is shown in Figure 5 (1–3 September 2015) with the 24 data points (8 each day) per product, to present a specific case of the diurnal and day-to-day variability for the $R_{rs}(\lambda)$ and the Chl-*a*, $a_g$(412) and POC products. The data were grouped by time of the day (color coded). In this particular case, the diurnal variability for all products is greater than the day-to-day variability for the individual local times for most times. For instance, for $R_{rs}$(412), the difference between the maximum and minimum for 1 September 2015 is greater than the difference between maximum and minimum among the three days (1–3 September 2015) for the 9:00 local time and the same applies for the rest of the times.

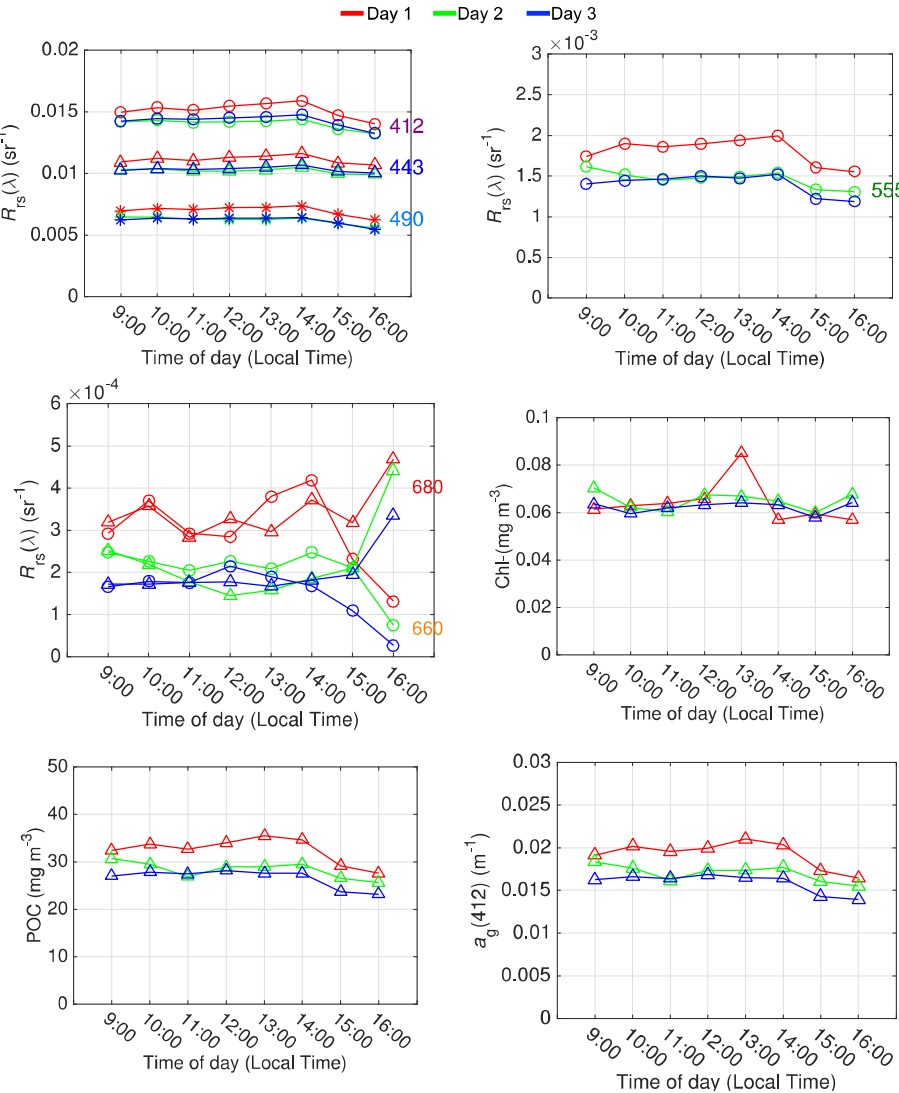

**Figure 5.** Three-day sequence for (**a**) $R_{rs}(\lambda)$ for 412, 443 and 490 nm bands, (**b**) $R_{rs}$(555), (**c**) $R_{rs}(\lambda)$ for 660 and 680 nm bands, (**d**) Chl-*a*, (**e**), $a_g$ (412) and (**f**) POC products. Data are color coded by day. Day 1, 2 and 3 = 1, 2 and 3 September 2015.

Various statistical parameters were calculated for all 96 three-day sequences in order to obtain a quantitative estimation of the diurnal variability for all cases. We use the percent coefficient of variation ($CV[\%] = 100 \times SD/\text{mean}$ with $SD$ the standard deviation) to describe the dispersion of the 24 values per sequence. Therefore, a $CV[\%]$ from 24 data points (3 days with 8 values per day) was calculated for all three-day sequences ($CV[\%]_{3\text{-day}}$) and therefore, we have 96 $CV[\%]_{3\text{-day}}$ values (Figure 6). A normality test was conducted to determine whether each 24-point data product for each 3-day sequence was normally distributed. A Kolmogorov-Smirnov test was performed to each dataset using the Matlab tool "kstest." The Kolmogorov-Smirnov test compares the dataset to a normal distribution with the null hypothesis that the dataset has a standard normal distribution. The null hypothesis is rejected if the test is significant at the 5% level. Most of the sequences passed the normality test and therefore, the CV is meaningful for these analyses. The sequences that failed to pass the test were not included in the analysis (shown as discontinuous lines in Figure 6a,c).

We observed that the mean (and median) for $CV[\%]_{3\text{-day}}$ is less than 10% for $R_{\text{rs}}$ at 412–555 and biogeochemical products (Figure 6; Table 1), even less than 5% for the $R_{\text{rs}}$ in the blue bands, indicating that the day-to-day variability (at least within the confines of the daytime period that GOCI observes) is small on average and therefore, demonstrating the homogeneity of the GCWS region. No information is available between the last GOCI observation of the day and first observation on the following day to evaluate variability during that time period.

Next, a diurnal mean and standard deviation ($SD_{\text{diurnal}}$) were calculated for each day (potentially 8 data points per day) for the whole GOCI mission. These values were calculated only if four or more values were valid per day. A normality test was also conducted on each 8-point dataset and all of them passed this test. Hence, we have confidence in applying the standard deviation and coefficient of variation for this analysis. Also, the diurnal percentage coefficient of variation ($CV[\%]_{\text{diurnal}}$) was calculated from these mean and values. Then, the mean of all the diurnal $SD$ values ($\overline{SD}_{\text{diurnal}}$) and the median of the percentage coefficient of variation ($\text{Med}[CV[\%]_{\text{diurnal}}$) were calculated for all the data (all seasons) and for summer alone, when the variability due to change in the in-water properties are minimal (Table 2). The $\text{Med}[CV[\%]_{\text{diurnal}}$ of $R_{\text{rs}}$ for the GCWS region is less than 5% for the blue and green bands and for the Chl-*a*, $a_{\text{g}}(412)$ and POC products for both all the seasons and only summer. These small $\text{Med}[CV[\%]_{\text{diurnal}}$ and mean of $CV[\%]_{3\text{-day}}$ support the assumption that the GCWS region is spatially and temporally homogeneous over the course of a day or day-to-day. Nevertheless, we acknowledge that the region of study also exhibits seasonality, as expected, which is reflected in the time series of Section 4.1 and very low-level of diurnal variability that cannot be discerned with GOCI and current processing capabilities. The significantly higher $\text{Med}[CV[\%]_{\text{diurnal}}$ values for the 660 and 680 nm bands are likely related to the low ocean reflectance signals at these red wavelengths in the GCWS. However, the 680 band could be expressing diurnal variability as discussed in material that follows.

**Table 1.** Statistics for the $CV[\%]_{3\text{-day}}$ of the biogeochemical products.

| | $CV[\%]_{3\text{-day}}$ | | | | | |
|---|---|---|---|---|---|---|
| **Product** | **min.** | **max.** | **mean** | **median** | **SD** | **N** |
| Chl-*a* | 3.01 | 24.80 | 9.17 | 7.72 | 4.74 | 93 |
| POC | 2.79 | 17.42 | 8.01 | 7.73 | 2.43 | 93 |
| $a_{\text{g}}(412)$ | 2.59 | 16.45 | 7.51 | 7.32 | 2.30 | 94 |

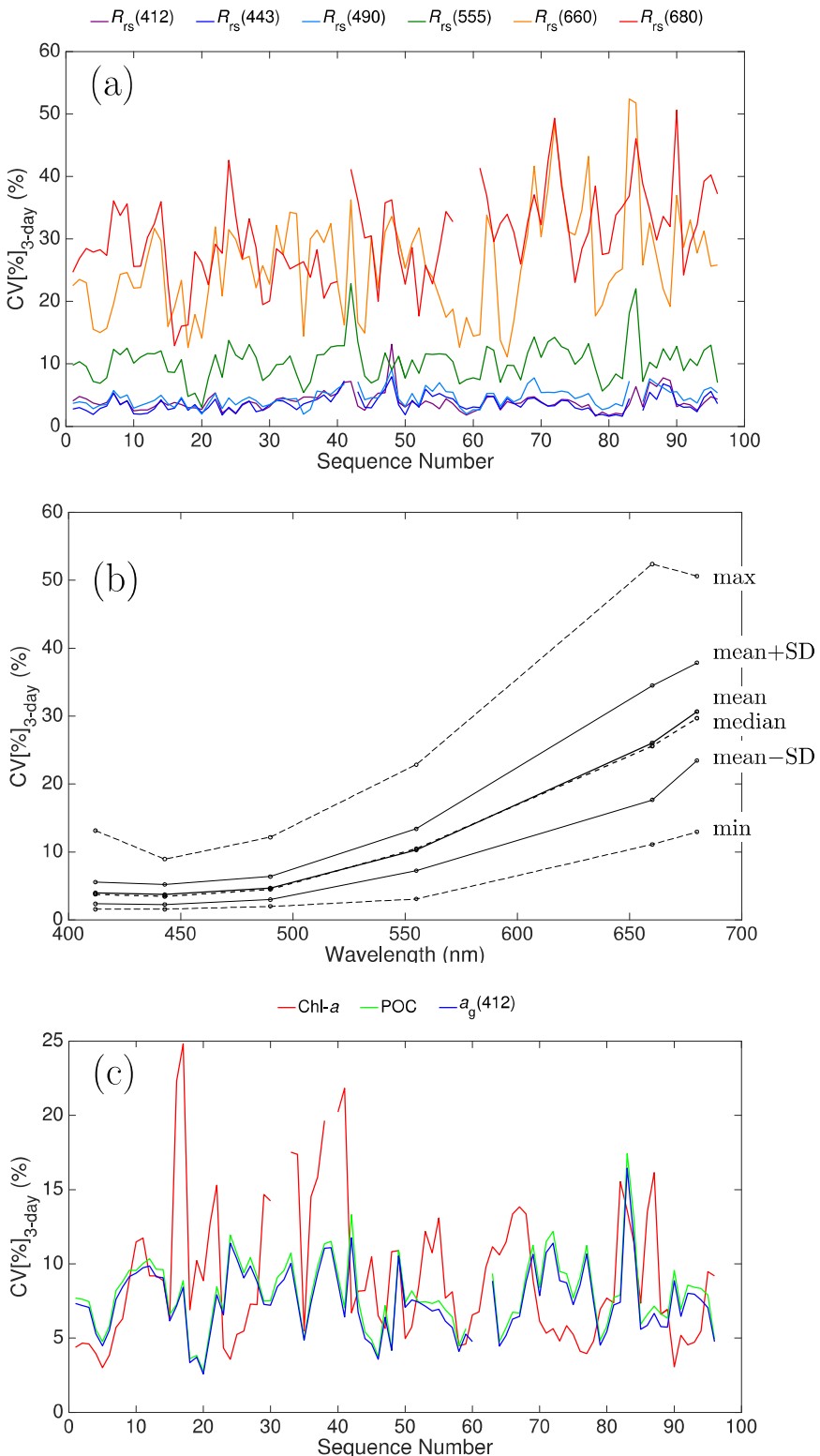

**Figure 6.** Diurnal variability in GOCI-derived (**a**) $R_{rs}$ by sequence number, (**b**) $R_{rs}$ by wavelength and (**c**) biogeochemical products demonstrated in terms of the coefficient of variation for each of the 96 three-day sequences of hourly observations ($CV[\%]_{3\text{-day}}$). Minimum (lower dashed line), maximum (upper dashed line), mean (solid bold black line), median (dashed bold black line) and $+/-1SD$ (solid thin line).

Additionally, the $\overline{SD}_{\text{diurnal}}$ is an indicator of the diurnal variability of the GCWS. Therefore, we consider that two times the $\overline{SD}_{\text{diurnal}}$ values (i.e., $2 \times \overline{SD}_{\text{diurnal}}$) for summer for the GCWS region, representing 95% of the normally distributed $R_{\text{rs}}$ values, provides an appropriate measure of the minimum $R_{\text{rs}}$ (or derived products) difference required to detect diurnal variability (Table 2). When compared with the RMSE from the $R_{\text{rs}}$ matchups between GOCI and AERONET-OC data [36], the $2 \times \overline{SD}_{\text{diurnal}}$ values are up to one order of magnitude smaller for nearly all bands (412–660 nm). The relatively higher AERONET-OC RMSE values can be attributed to in situ measurement uncertainties and the proximity of these sites to land contributing to higher uncertainties in the atmospheric correction from more complex aerosol constituents and absorbing trace gases (ozone and nitrogen dioxide) and more optically complex water types (i.e., higher sediment, biogenic particles and CDOM) in both time and space.

**Table 2.** Measures of the diurnal variance for GOCI $R_{\text{rs}}$ and derived biogeochemical products in the GCWS study region during summer and all seasons combined. Two times the mean of the diurnal SD ($2 \times \overline{SD}_{\text{diurnal}}$) for summer (in bold) is considered the threshold uncertainty associated with the GOCI sensor. The median of the percentage coefficient of variation ($\text{Med}[CV[\%]_{\text{diurnal}}$) of the diurnal values and the root mean squared error (RMSE) from the AERONET-OC data (Concha et al. 2019) [36] are shown for reference. $R_{\text{rs}}(\lambda)$ in units of $\text{sr}^{-1}$, Chl-*a* in units of mg m$^{-3}$, $a_{\text{g}}(412)$ in units of m$^{-1}$ and POC in units of mg m$^{-3}$.

| Product | All Seasons | | | Summer | | | AERONET-OC |
|---|---|---|---|---|---|---|---|
| | $2 \times \overline{SD}_{\text{diurnal}}$ * | $\text{Med}[CV[\%]_{\text{diurnal}}$ ** | N | $2 \times \overline{SD}_{\text{diurnal}}$ * | $\text{Med}[CV[\%]_{\text{diurnal}}$ ** | N | RMSE * |
| $R_{\text{rs}}(412)$ | $1.08 \times 10^{-3}$ | 3.90 | 1160 | $8.05 \times 10^{-4}$ | 2.60 | 403 | $2.2 \times 10^{-3}$ |
| $R_{\text{rs}}(443)$ | $7.10 \times 10^{-4}$ | 3.32 | 1160 | $5.49 \times 10^{-4}$ | 2.32 | 403 | $1.8 \times 10^{-3}$ |
| $R_{\text{rs}}(490)$ | $5.40 \times 10^{-4}$ | 3.85 | 1160 | $4.48 \times 10^{-4}$ | 2.98 | 403 | $2.1 \times 10^{-3}$ |
| $R_{\text{rs}}(555)$ | $2.77 \times 10^{-4}$ | 7.57 | 1160 | $2.51 \times 10^{-4}$ | 6.72 | 403 | $2.3 \times 10^{-3}$ |
| $R_{\text{rs}}(660)$ | $9.68 \times 10^{-5}$ | 20.19 | 1159 | $8.83 \times 10^{-5}$ | 16.85 | 403 | $5.0 \times 10^{-4}$ |
| $R_{\text{rs}}(680)$ | $1.08 \times 10^{-4}$ | 17.63 | 1159 | $1.36 \times 10^{-4}$ | 20.40 | 403 | N/A |
| Chl-*a* | $1.57 \times 10^{-2}$ | 6.15 | 1155 | $1.09 \times 10^{-2}$ | 5.71 | 401 | N/A |
| $a_{\text{g}}(412)$ | $2.26 \times 10^{-3}$ | 4.52 | 1159 | $2.09 \times 10^{-3}$ | 5.12 | 402 | N/A |
| POC | 4.03 | 4.91 | 1159 | **3.70** | 5.37 | 402 | N/A |

\* $R_{\text{rs}}$ in sr$^{-1}$, $a_{\text{g}}(412)$ in m$^{-1}$, and Chl-*a* and POC in mg m$^{-3}$. ** in [%].

In order to determine what time of the day the variability in GOCI $R_{\text{rs}}$ and biogeochemical data products is greater and whether these are significant, the relative difference of the time of the day, $t$, with respect to the value at 13:00 hours ($R\Delta_t[\%]$) was calculated (Table 2; Figure 7). The value at 13:00 hours was chosen as a reference because it reflects the value that NASA heritage sensors (SeaWiFS, MODIS-Aqua, VIIRS) would measure with similar acquisition time and solar geometry. The values at this time of day should be affected less by solar geometry (lower solar zenith angle) than the early and late periods of the day. Assuming temporal homogeneity, we would expect minimal deviation from the value at 13:00 hours.

If we define the difference with respect to a reference as $\Delta_t = x_t - x_{\text{reference}}$, then, the relative difference for time $t$ is defined as

$$R\Delta t[\%] = \frac{\Delta_t}{|x_{\text{reference}}|} \times 100[\%] = \frac{x_t - x_{\text{reference}}}{|x_{\text{reference}}|} \times 100[\%] \qquad (7)$$

where $x_t$ is the satellite data at the local time $t$ = 09:00, 10:00, ..., 16:00 hours and in this case, the reference is the value at 13:00 hours, that is, $x_{\text{reference}} = X_{13:00}$. The $R\Delta_t[\%]$ is an indicator of uncertainties that are expected depending on the time of the day, assuming no changes in the in-water properties due to biological or biogeochemical processes. Figure 7 shows a heat map for $R\Delta_t[\%]$ of GOCI-derived products color coded by the number of points (frequency) that fall within the different $R\Delta_t[\%]$ intervals in the *y*-axis. The x-axis represents the time of day. Overall, most of the $R\Delta_t[\%]$ values fall below 10% for all bands except the 660 and 680 nm bands. Most of the $R\Delta_t[\%]$ values are below or close to 5% for

the blue bands for all times of the day. For the 660 nm band, the mean $R\Delta_t[\%]$ values for the last two times of day are about 20%. For the 680 nm band, most of the $R\Delta_t[\%]$ values fall within 40% for all times of the day except the last two values at the end of the day. This relative difference in $R_{rs}(680)$ could be attributed to diurnal variability in solar-induced phytoplankton fluorescence. O'Malley et al. (2014) [48] found evidence of diurnal variability in non-photochemical quenching (NPQ) in GOCI observations expressed as higher chlorophyll-*a* normalized fluorescence signal before and after the mid-day period due to an alleviation from the peak in phytoplankton NPQ occurring during the mid-day period when light intensity is maximal. These results suggest that the diurnal fluorescence response may be sufficiently high in the GCWS to exceed GOCI sensor uncertainty in the $R_{rs}(680)$ band. This effect may be also influencing the higher CV values in Figure 6 and higher $R_{rs}(680)$ at 16:00 in Figure 5c. For Chl-*a*, POC and $a_g(412)$ for all times of the day, $R\Delta_t[\%]$ are less than 10% (Table 2; Figure 7).

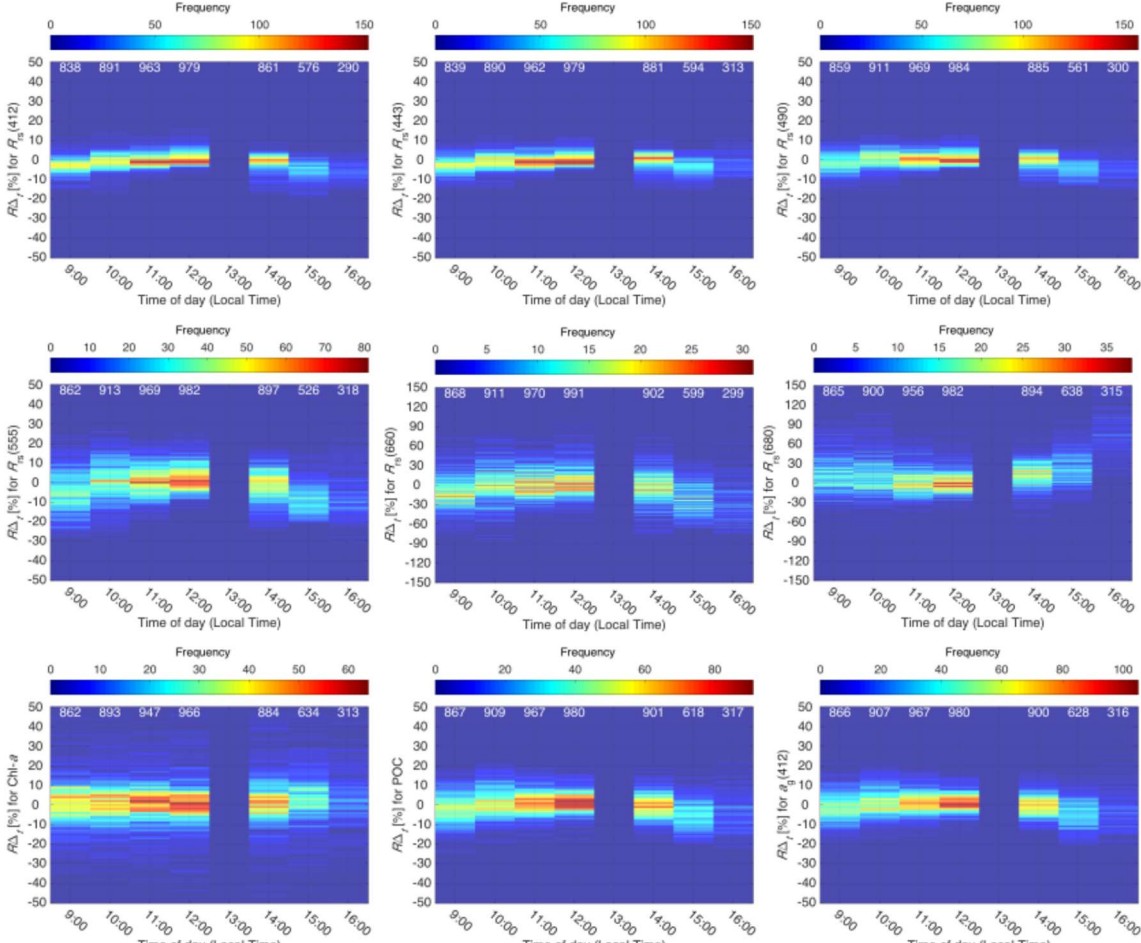

**Figure 7.** Relative difference $R\Delta_t[\%]$ with respect to the value at 13:00 hours for (**a**–**f**) $R_{rs}(\lambda)$ and (**g**) Chlorophyll-*a*, (**h**) POC and (**i**) $a_g(412)$. The color corresponds to the number of $R\Delta_t[\%]$ values (or frequency) to demonstrate the dispersion of $R\Delta_t[\%]$ by hour of observation for each data product. The sample size for each time of day are indicated in the top of the figure (white font).

The variability in $R_{rs}(\lambda)$, Chl-*a*, $a_g(412)$ and POC products versus SZA was investigated to evaluate the extent to which imperfect atmospheric correction and BRDF models due to elevated SZA factors, such as higher air mass fraction and lower signal, affect the uncertainty in product retrievals. This analysis implicitly examines the complete Sun-viewing geometry through the seasons. GOCI data from the summer period show the lowest level of variability for all hourly observations (Figure 8). Overall, $R_{rs}$ and other products do not seem to be negatively affected by SZA to any appreciable level and no

negative or invalid values are recorded at extreme SZA (SZA > 75°) for most of the products except the red bands. The negative values seem to occur at SZA > 75°. This indicates that the atmospheric correction model seems to adequately account for solar geometry effects, even at relatively high SZA.

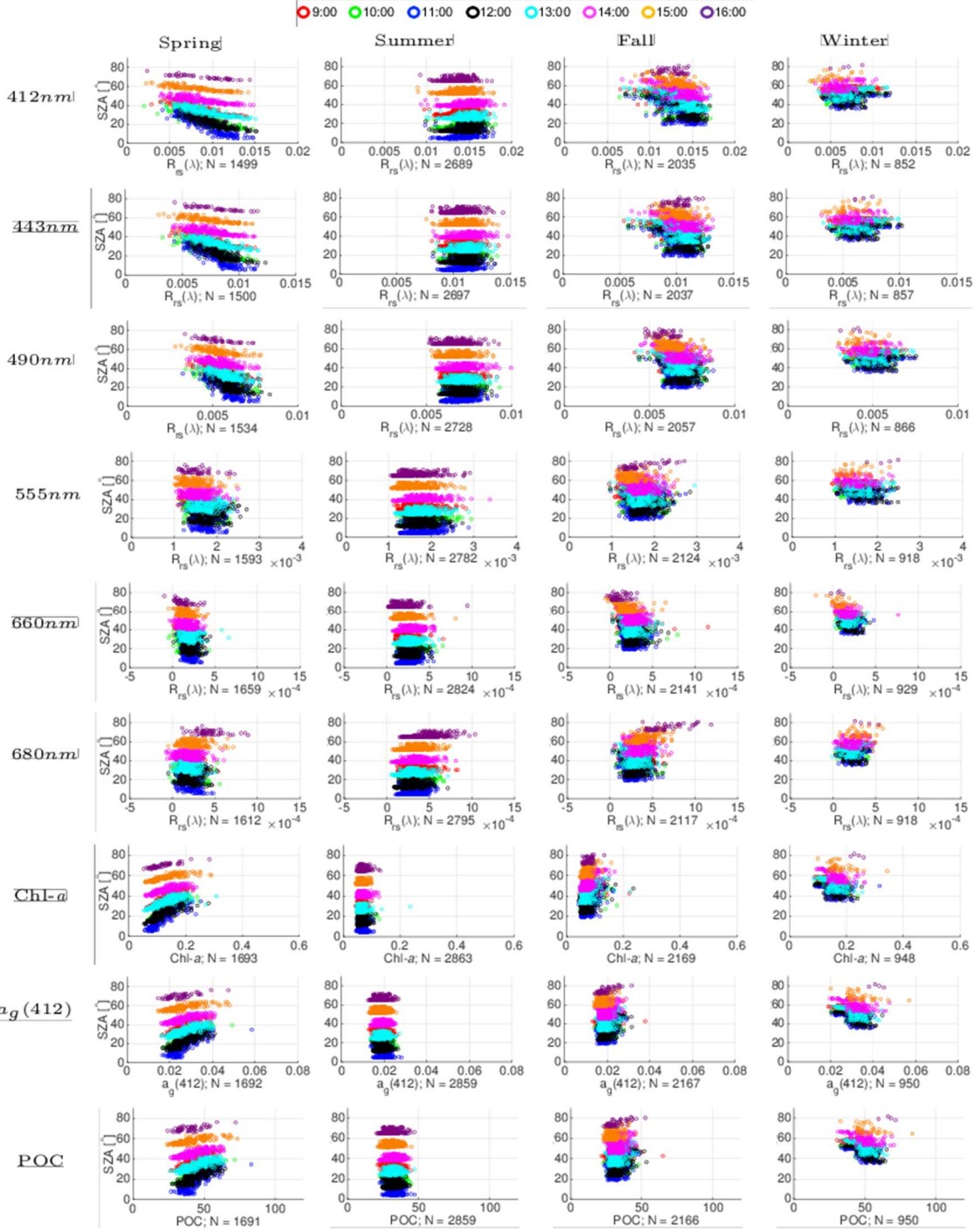

**Figure 8.** Filtered mean for $R_{rs}(\lambda)$ and biogeochemical products versus solar zenith angle (SZA) for the GOCI mission from May 2011 to January 2018. Only data that passed the exclusion criteria were used. All SZA values (0° < SZA < 90°) were used. The data are separated by season and color coded by time of the day.

Summer is fairly uniform with very narrow ranges of variability for each product, indicating that the in-water properties are fairly stable during summer regardless of the SZA. There is a wider

range of values during the other seasons, especially in spring due to the higher productivity yielding a greater amplitude in $R_{rs}$ and biogeochemical products (Figure 8). The greatest number of valid values is from summer, followed by fall, then spring, being lowest in winter. In spring and fall, there are almost twice as many valid values than for winter and three times for summer. For winter, there are more limited observations due to the quality screening criteria excluding data as well as a wider range of values at higher SZA. Similarly to $R_{rs}$, the derived biogeochemical products demonstrated a wide range of values during spring and narrower distribution during summer.

## 5. Summary and Conclusions

The GOCI mission times series (May 2011 to January 2018) of hourly $R_{rs}$ and biogeochemical products (Chl-*a*, CDOM absorption at 412 nm and particulate organic carbon) was investigated for a region of assumed homogenous in-water optical properties with the objective of estimating threshold GOCI $R_{rs}$ retrieval uncertainties to enable studies of diurnal variability in more dynamic regions. With the possible exception of the phytoplankton fluorescence signal in $R_{rs}(680)$, the remaining GOCI $R_{rs}$ and biogeochemical products studies demonstrated no measurable sub-diurnal to day-to-day variability of in-water properties in the GCWS. While oligotrophic regions such as the GCWS do undergo diurnal variability, the extent of such biogeochemical processes cannot be discerned with GOCI and ocean color processing capabilities at this time. An expected seasonal cycle was observed through the entire mission for all products (Figures 3 and 4). No negative values were obtained for the $R_{rs}$ products for most bands except for a few negative values in the red bands, which demonstrates that the atmospheric correction and vicarious calibration are performing adequately. This study employed the new vicarious gains by [36] for NASA's processing of GOCI data, which improved the quality of the $R_{rs}$ products dramatically from the previous NASA processing.

We consider that an approximate measure of the threshold or minimum difference required for $R_{rs}$ (or derived products) to detect diurnal or day-to-day change in GOCI is considered to be two times the mean diurnal $SD$ values (i.e., $2 \times \overline{SD}_{diurnal}$) for summer for the GCWS region (Table 2). This estimation of variability was determined from summer because in this season there is less variability due to change in the in-water properties (Figure 8). The $2 \times \overline{SD}_{diurnal}$ values are at least three times smaller for all bands (412–660 nm) when compared with the RMSE from the matchups from AERONET-OC data. The RMSE values from AERONET-OC are an estimation of the uncertainties for more productive waters and under more challenging atmospheric conditions, while $2 \times \overline{SD}_{diurnal}$ provide uncertainties levels under more constant oceanic atmospheric conditions. Therefore, the changes in the in-water properties retrieved by GOCI should at least be greater than these threshold values to be considered a change in water properties due to biological, physical or biogeochemical processes.

Overall, the relative difference with respect to the value at 13:00 hours ($R\Delta_t[\%]$) are less than 10% for all products except $R_{rs}(660)$ and $R_{rs}(680)$ and there is no indication of skewing of the diurnal variability due to processing (Table 2; Figure 7). A similar behavior occurs for the other biogeochemical products examined, with a relative difference less than 10% for all times of the day. It appears to be a small effect from SZA at the beginning and end of the day presumably due to reduced amount of sunlight and thus reduced SNR of the sensor. However, the effect of SZA is very minimal and no significant trend is observed. When the GOCI data versus SZA were analyzed (Figure 8), separated by seasons and by time of day, no trend was observed. Summer seems to yield the most ideal data for evaluating GOCI in this study area because of the very narrow variability for all products. Spring is more variable in the Chl-*a* levels, as a consequence of phytoplankton production (Figure 8). The same behavior can be seen for $a_g(412)$ and POC. From these two analyses, it seems that the time of day and therefore the SZA, does not have a significant negative impact on the results that passed the filtering criteria, demonstrating that the atmospheric correction algorithm is working adequately, even at extreme SZA (SZA > 75°). It is worth pointing out that this holds true only if a proper vicarious calibration is applied. Before updating the vicarious gains, low or negative $R_{rs}$ values were observed at extreme SZA.

We acknowledge the fact that some diurnal variability occurs in the GCWS due to biogeochemical or biological processes and these changes could be embedded in our results. However, we believe that this variability is minimal, especially in summer and the GCWS is homogeneous in time and space with no diurnal trend that can be discerned through GOCI observations.

As a general conclusion, the diurnal variability estimates determined in this study provide a guide as to the minimum value of diurnal change that must be observed to overcome uncertainties in instrument radiometric noise and algorithm processing. Our future work will apply these results to estimate changes in diurnal and day-to-day biogeochemical stocks and processes in the coastal ocean using GOCI.

**Author Contributions:** Conceptualization, J.C. and A.M.; Methodology, J.C. and A.M.; Software, J.C.; Validation, J.C.; Formal Analysis, J.C.; Investigation, J.C.; Resources, J.C, A.M., B.F. and W.K.; Data Curation, J.C.; Writing-Original Draft Preparation, J.C.; Writing-Review & Editing, J.C, A.M., B.F. and W.K.; Visualization, J.C.; Supervision, A.M.; Project Administration, A.M.; Funding Acquisition, A.M. and B.F.

**Funding:** NASA Project ROSES Earth Science U.S. Participating Investigator (NNH12ZDA001N-ESUSPI).

**Acknowledgments:** We acknowledge that GOCI L1B data in OBPG are from Korea Institute of Ocean Science and Technology (KIOST) coordinated with Korea Ministry of Oceans and Fisheries (MOF).

**Conflicts of Interest:** The authors declare no conflicts of interest.

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
