# Peer review of "Uncertainties in the Geostationary Ocean Color Imager (GOCI) Remote Sensing Reflectance for Assessing Diurnal Variability of Biogeochemical Processes"

_remotesensing, doi:10.3390/rs11030295_

Round 1
Reviewer 1 Report
General comment:
Concha et al. analyzed the multi-year GOCI satellite Rrs measurements and evaluated the diurnal variability of Rrs as well as some bio-optical products derived from GOCI. They derived the first sets of base values or detection threshold for water properties. Overall, it is well written and easy to follow. The results represent a novel contribution to the ocean color community and can be used for further study on the diurnal variability of coastal environments. So I would recommend for publication. I only have a few minor specific questions.
Specific questions:
Line 257: … used to test the homogeneity
Line 268: You compared with AERONET-OC data. But I did not find that reference... So I would suggest you to provide a bit more details.
Figure 6: Caption: “Minimum” and “maximum” should be corrected.
Line 332: I think there are negative values in your Rrs products.
Last, I have some concerns about the base values derived for Chl-a, ag(412) and POC, as the secondary uncertainties from their inversion algorithms are involved. It is definitely worth a thorough discussion.
Reviewer 2 Report
Please see attachment.

Reviewer 3 Report
In general, it is a very useful and important study, specially with the plans for new geostationary sensors. Also, it can give insights for other types of sensors uncertainties analysis. The time-series analyzed in this study provide very good idea of the uncertainties that may be applied on the products retrievals, although it is limited to this regional sensor. I suggest that the results found here could be further investigated and discussed in terms of sources of uncertainties for this specific sensor, but also how it compares to the uncertainties from other ocean color sensors. Furthermore, the authors could provide a suggestion on how to use those findings to improve the products retrievals from this sensor for studies that aim to detect short-time variability of biogeophysical variables using this approach. It would be very interesting to see an implementation of the uncertainties proposed here for an area covered by the sensor, especially if the authors could include any validation of the retrievals or radiometry.

Round 2
Reviewer 3 Report
I found this version of the work significantly better and one of my main concerns were addressed and discussed. I have still very minor comments in the file attached.

Author Response
I would suggest to remove “Threshold” from the title. Only a suggestion here though.
Response:
Good suggestion. Changed.
Lines 57-58: there is some misspelling here.
Response:
Fixed.
Area of study - I still have a hard time to understand why you present the GCWS area, assuming that this is a region with small short-term variability, but in the next paragraph, you describe all the variability that happens in this region as a whole. Would it make sense if you first introduce the region, with all the oceanographic dynamics, and then you introduce the GCWS area, and explain why that subset was chosen, despite all the variability you described? Maybe only switching the paragraphs and explaining that “despite this dynamics, the GCWS region was selected because..”?
Response:
Both paragraphs were moved around and edited following the reviewer suggestion.
Results and conclusions:
Do you think that another season, like winter, would be a better period to test the uncertainties of bands 660 and 680nm, because of the influence of high light and NPQ effect?
Response:
This is a good suggestion. However, for the purpose of this work, we decided to use the same season for all products to maintain consistency.
I think that including the discussion about the diurnal fluorescence signal detected by the GOCI was really essential in this discussion.
Response:
Agree. Thanks for the suggestion.
I think lines 406 to 409 could be omitted from conclusions section.
Response:
Lines were deleted.
